# Comparability of calibration strategies for measuring mercury concentrations in gas emission sources and the atmosphere

Iris de Krom[1], Wijnand Bavius[1], Ruben Ziel[1], Elizabeth A. McGhee[2], Richard J. C. Brown[2], Igor Živković[3], Jan Gačnik[3], Vesna Fajon[3], Jože Kotnik[3], Milena Horvat[3] and Hugo Ent[1]

[1] VSL, Department of Chemistry, Mass, Pressure and Viscosity, Thijsseweg 11, 2629 JA Delft, the Netherlands
[2] Environment Department, National Physical Laboratory, Hampton Road, Teddington, TW11 0LW, UK
[3] Jozef Stefan Institute, Department of Environmental Sciences, Jamova Cesta 39, 1000 Ljubljana, Slovenia

*Correspondence to*: Iris de Krom (idekrom@vsl.nl)

**Abstract.** A primary mercury gas standard was developed at VSL to establish an SI-traceable reference point for mercury concentrations at emission and background levels in the atmosphere. The majority of mercury concentration measurements are currently made traceable to the empirically determined vapour pressure of mercury. The primary mercury gas standard can be used for the accurate and precise calibration of analytical systems used for measuring mercury concentrations in air. It has been especially developed to support measurements related to ambient air monitoring (1 ng m$^{-3}$ – 2 ng m$^{-3}$), indoor and workplace related mercury concentration levels according to health standards (from 50 ng m$^{-3}$ upwards) as well as to stationary source emissions (from 1 µg m$^{-3}$ upwards).

The primary mercury gas standard is based on diffusion according to ISO 6154-8. Calibration gas mixtures are obtained by combining calibrated mass flows of nitrogen and air through a generator holding diffusion cells, containing elemental mercury. In this paper, we present the results of comparisons between the primary gas standard and mercury calibration methods maintained by NPL, a National Metrology Institute (NMI), and JSI, a Designated Institute (DI). The calibration methods currently used at NPL and JSI are based on the bell-jar calibration apparatus in combination with the Dumarey equation or a NIST reference material. For the comparisons, mercury was sampled on sorbent traps to obtain transfer standards with levels between 2 ng and 1000 ng with an expanded uncertainty not exceeding 3 % ($k = 2$). The comparisons performed show that the results for the primary gas standard and the NIST reference material are comparable, whereas a difference of -8 % exists between results traceable to the primary gas standard and the Dumarey equation.

## 1. Introduction

Mercury is a global pollutant and in its many chemical forms highly toxic to human, animal and environmental health. Mercury occurs naturally in the environment, and in addition human activities have increased the atmospheric mercury concentration up to 500 % above natural levels (Global mercury assessment 2018). Reliable and comparable measurement results of mercury

concentration levels in the environment are key to underpin global efforts to control and reduce the mercury emissions, to meet the obligations of legislation and to protect human health.

The majority of measurements of mercury concentrations are currently traceable to the empirically determined equations describing the saturated vapour pressure of mercury, usually via a bell-jar calibration apparatus (Brown et al., 2008a, Brown et al., 2008b). This apparatus allows a saturated concentration of mercury vapour in air to develop in a confined space in equilibrium with ambient conditions, from which a known amount of mercury can be removed for calibration purposes. Several empirical equations are available, e.g., the Dumarey and the, more recently proposed, Huber equations (Dumarey et al., 1979, Dumarey et al., 1985, Dumarey et al., 2010, Huber et al., 2006). The agreement between these equations is unsatisfactory, with vapour pressures from different equations differing by more than 7 % at 20 ºC (Brow et al., 2010, Quétel et al., 2014). To remove the dependency of mercury concentration measurements from these empirical equations, and to provide stability and comparability, a primary mercury gas standard has been developed by VSL (Van Swinden Laboratory, National Metrology Institute in the Netherlands), to provide metrological traceability to the International System of Units (SI) (Ent et al., 2014, de Krom et al., 2020). The SI-traceable primary gas standard, working according to ISO 6145-8 (ISO 6145-8, 2005), can be used for the calibration of instruments and the certification of mercury gas generators. Furthermore, sorbent traps can be spiked with known amounts of mercury from the primary gas standard using pumped sampling, according to ISO 16017-1 (ISO 16017-1, 2000). The transfer standards obtained can be used as certified reference materials to calibrate field instruments or for comparisons such as mandatory for calibration and testing laboratories to show their conformity assessment under ISO/IEC 17025 (ISO/IEC 17025, 2017). Such a comparison can only be successful if it allows participants to validate their measurement procedures with respect to SI-traceable standards as required by ISO/IEC 17043 (ISO/IEC 17043:2010).

Measurements of mercury concentrations in air are typically carried out using a pump to sample air at a monitoring location, at a known rate for a known time, onto an adsorption tube (Pirrone et al., 2013, Pandey et al., 2011). The trapping of mercury is usually performed with a sorbent tube using gold to form an amalgam and trap the mercury. To increase the surface area available, the gold is often dispersed on silica or a similar support (Brown et al., 2011, Brown et all., 2017). Mercury in this adsorption tube is then thermally desorbed and measured. Furthermore, in 2019 a new technical specification was adopted in Europe, which describes the use of gold amalgamation traps for sampling and determination of mercury compounds in flue gas (NVN-CEN/TS 17286, 2019). This technical specification is based on the United States Environmental Protection Agency (US EPA) 30B method (Method 30B, 2017). That method in turn uses carbon sorbent traps for the determination of total gaseous mercury emissions from coal-fired combustion sources. The sorbent traps contain carbon typically activated with iodine or another halogen (Živković et al., 2020).

To demonstrate the robustness and comparability of transfer standards obtained with the primary gas standard, in this paper we present the results of comparisons with current calibration methods maintained, using the bell-jar in combination with the Dumarey equation or the NIST (National Institute of Standards and Technology in the United States) liquid standard reference

material (SRM 3133) for calibration of the equipment. For the comparisons, gold sorbent tubes and carbon sorbent tubes were sampled with the primary mercury gas standard to obtain transfer standards. For the first comparison, between NPL (The National Physical Laboratory in the United Kingdom) and VSL, gold sorbent tubes have been sampled with mercury amounts between 2 ng and 10 ng. Calibration of the analysers were performed with injections from the bell-jar in combination with the Dumarey equation. During the second comparison, between JSI (Jozef Stefan Institute in Slovenia) and VSL, amounts between 10 ng and 1000 ng have been sampled and the NIST SRM 3133 has been used to calibrate the analyser used.

## 2. Materials and methods

### 2.1 Primary mercury gas standard

The primary gas standard has been developed as an elemental mercury ($Hg^0$) gas generator to establish metrological traceability of mercury concentration measurement results, based upon a gravimetric approach, for ambient air levels as well as higher concentrations (Ent et al., 2014, de Krom et al., 2020) .

The working principle of the primary mercury vapour generator is based on diffusion according to ISO 6145-8 (ISO 6145-8, 2005). Specially developed stainless-steel diffusion cells are filled with approximately 2 mL of $Hg^0$. The $Hg^0$ vapour diffuses from the cell through a capillary. To obtain several diffusion ranges, cells with different capillary diameters are used, e.g.: 33 mm and 3 mm. By weighing the diffusion cells at regular time intervals on a high-resolution analytical balance (AX1006 mass comparator with a Mettler AT1005 balance, Mettler, Switzerland) the $Hg^0$ mass flow rate (diffusion) is determined gravimetrically. The characterisation of different diffusion cells has been described previously by I. de Krom et al (de Krom et al., 2020).

In the dynamic mercury gas generator, the cells are housed in a diffusion chamber. The diffusion chamber is temperature (20.0 ºC $\pm$ 0.1 ºC) and pressure (105.0 kPa $\pm$ 0.1 kPa) controlled, At the bottom a nitrogen flow of 500 mL min$^{-1}$ enters the diffusion chamber. All of the flow, also enriched by mercury vapour, is then guided out of the diffusion chamber through an aperture at the top. Standard $Hg^0$ gas mixture concentrations are prepared by mixing the $Hg^0$ vapours in nitrogen with flows, between 1 L min$^{-1}$ and 20 L min$^{-1}$, of matrix gas, e.g., purified air. Using diffusion cells with a capillary of 3 mm in diameter, mercury concentrations between 0.1 µg m$^{-3}$ and 2.1 µg m$^{-3}$ can be obtained with an expanded uncertainty of 3 %. Diffusion cells with a capillary of 33 mm in diameter can generate mercury concentrations between 5 µg m$^{-3}$ and 100 µg m$^{-3}$ with an uncertainty of 1.8 %. In this project both types of diffusion cell have been used to obtain the primary mercury gas standard.

### 2.2 Preparation of transfer standards

Transfer standards are prepared via pumped sampling of known volumes of the primary mercury gas standard for a known time onto sorbent traps according to ISO 16017-1 (ISO 16017-1, 2000), using a specially designed multi-sampling manifold,

made up of 6 thermal Mass Flow Controllers (MFCs) operating in sucking mode. Each MFC is connected to a 3-way valve. All valves are controlled with a timer, that allows them to switch simultaneously. The sampling is done under controlled environmental conditions.

The mercury amount collected onto the sorbent material can be calculated using equation (1).

$$m_{\text{Hg}} = \frac{\bar{q}_m(\text{Hg})\, t\, q_v(\text{sample})}{q_v(\text{air})} \tag{1}$$

$m_{\text{Hg}}$ is the mercury amount on the sorbent material in ng, $\bar{q}_m(\text{Hg})$ is the mercury diffusion rate from the diffusion cell(s) in ng min$^{-1}$, $t$ is the sampling time in minutes, $q_v(\text{sample})$ is the pumped sampling flow in L min$^{-1}$ and $q_v(air)$ is the total gas flow in L min$^{-1}$ through the generator to obtain the primary mercury gas standard. As an example, the standard measurement uncertainty associated with a mercury amount of 10.5 ng, obtained using three diffusion cells with a capillary of 3 mm in diameter, can be calculated from equation (1) using the law of propagation of uncertainty of the Guide to the expression of Uncertainty in Measurement (GUM) (BIPM, 2008). In Table 1 the uncertainty budget belonging to the mercury amount is shown. The expanded uncertainty is 0.3 ng ($k = 2$), which is equivalent to a relative expanded uncertainty of 3%.

Table 1: Uncertainty budget for mercury amount sampled onto sorbent materials ($k = 1$) calculated from equation (1) using the law of propagation of uncertainty according to the GUM

| Measurand | Value | Distribution | Standard uncertainty | Sensitivity coefficient | Uncertainty contribution |
|---|---|---|---|---|---|
| $\bar{q}_m$(Hg cell 18) | 1.37 ng min$^{-1}$ | Normal | 0.03 ng min$^{-1}$ | 2.490 min | 0.075 ng |
| $\bar{q}_m$(Hg cell 19) | 1.44 ng min$^{-1}$ | Normal | 0.04 ng min$^{-1}$ | 2.490 min | 0.110 ng |
| $\bar{q}_m$(Hg cell 20) | 1.41 ng min$^{-1}$ | Normal | 0.04 ng min$^{-1}$ | 2.490 min | 0.090 ng |
| $t$ | 75.000 min | Normal | 0.015 min | 0.140 ng min$^{-1}$ | 0.002 ng |
| $q_v(sample)$ | 0.1002 L min$^{-1}$ | Normal | 0.0004 L min$^{-1}$ | 150.094 ng L$^{-1}$ min | 0.037 ng |
| $q_v(air)$ | 3.018 L min$^{-1}$ | Normal | 0.008 L min$^{-1}$ | -3.489 ng L$^{-1}$ min | -0.029 ng |
| **$m_{\text{Hg}}$** | **10.5 ng** | **Normal** | **0.17 ng** | | |

## 2.3 First comparison: 2 – 10 ng mercury

Amasil sorbent tubes (PS Analytical, UK) with a specific area of 100 m$^2$ g$^{-1}$ have been used in this study. The sorbent material used in these tubes is gold-coated silica. The primary gas standard is obtained using two diffusion cells with a capillary of 3 mm in diameter (diffusion rate of 2.92 ng min$^{-1}$ ± 0.09 ng min$^{-1}$ ($k = 2$)). Different system flow rates are used to obtained different mercury concentrations (x(Hg)) (Table 2). The sorbent tubes were sampled, during three different rounds (in May, July and September 2017), with 5 different amounts (approximately 2 ng, 4 ng, 6 ng, 8 ng and 10 ng) using a sample flow of

0.5000 L min$^{-1}$ ± 0.0018 L min$^{-1}$ ($k = 2$) for all sorbent tubes. During round 1 (May 2017), 5 sorbent tubes per amount were

sampled simultaneously, 2 sorbent tubes were analysed by NPL and 3 by VSL. Thereafter, 2 tubes per amount were sampled simultaneously in round 2 (July 2017) and round 3 (September 2017) which were analysed by NPL (Table 2). Both laboratories calibrated the analyser using injections from a bell-jar in combination with the Dumarey equation.

**Table 2: Parameters used to sample sorbent tubes from the primary mercury gas standard for the first comparison**

| Round | System flow rate (L min$^{-1}$) (U ($k = 2$)) | x(Hg) (µg m$^{-3}$) (U ($k = 2$)) | Sampling time (min) (U ($k = 2$)) | Mercury amount per tube (ng) (U ($k = 2$)) |
|---|---|---|---|---|
| 1 | 20.01 (0.12) | 0.146 (0.04) | 28.00 (0.03) | 2.04 (0.06) |
| | 10.03 (0.06) | 0.291 (0.09) | 28.00 (0.03) | 4.07 (0.13) |
| | 10.00 (0.06) | 0.292 (0.09) | 42.00 (0.03) | 6.12 (0.19) |
| | 5.901 (0.017) | 0.494 (0.15) | 33.50 (0.03) | 8.28 (0.26) |
| | 5.901 (0.017) | 0.494 (0.15) | 42.00 (0.03) | 10.38 (0.32) |
| 2 | 14.98 (0.09) | 0.195 (0.06) | 21.00 (0.03) | 2.04 (0.06) |
| | 10.00 (0.06) | 0.292 (0.09) | 28.00 (0.03) | 4.08 (0.13) |
| | 10.00 (0.06) | 0.292 (0.09) | 42.00 (0.03) | 6.12 (0.19) |
| | 6.022 (0.017) | 0.484 (0.15) | 34.00 (0.03) | 8.23 (0.26) |
| | 6.022 (0.017) | 0.484 (0.15) | 42.00 (0.03) | 10.17 (0.32) |
| 3 | 15.04 (0.09) | 0.194 (0.06) | 21.00 (0.03) | 2.04 (0.06) |
| | 9.99 (0.06) | 0.292 (0.09) | 28.00 (0.03) | 4.08 (0.13) |
| | 9.99 (0.06) | 0.292 (0.09) | 42.00 (0.03) | 6.13 (0.19) |
| | 5.995 (0.017) | 0.486 (0.15) | 34.00 (0.03) | 8.26 (0.26) |
| | 6.021 (0.017) | 0.484 (0.15) | 42.00 (0.03) | 10.16 (0.32) |

Analysis of samples took place at NPL using a PS Analytical Sir Galahad II analyser (PS Analytical, UK) with a fluorescence detector, using NPL's procedure, accredited by UKAS to ISO/IEC 17025 (ISO/IEC 17025, 2017), which is in accordance with the published reference method EN 15852(EN15852, 2010) (NPL's manual variant of EN 15852 has been shown to be equivalent to the automatic reference method within the uncertainty of the analytical determination (Brown et al., 2012)). The

instrument was calibrated prior to analysis using a gas-tight syringe, making multiple injections of known amounts of mercury vapour from the bell-jar onto the permanent trap of the analyser, across the range of expected sample concentrations. Sampled adsorption tubes were placed in the remote port of the instrument and heated to 900 $^{\circ}$C, desorbing the mercury onto a permanent trap. Subsequent heating of this trap then desorbed the mercury onto the detector for final measurement. Samples are desorbed three times to ensure all the mercury has been removed. It is assumed that the third desorption is equal to the blank level of the

tube and this response is taken off the response of the first and second desorption, which are then added together to provide a total analytical response. Quality control injections are made in-between samples to ensure that the analyser is not drifting outside a specified range. A conservative estimate of the relative expanded analysis uncertainty is between 8 % and 10 %. The main components of this uncertainty come from the uncertainty in the response from the adsorption tube during desorption, the repeatability of the instrument response and residual instrument drift (Brown et al., 2008c).

At VSL a PS Analytical 10.525 Sir Galahad (PS Analytical, UK) is used to analyse the sorbent tubes. For calibration a commercially available bell-jar has been used for comparison between the primary mercury gas standard and the Dumarey equation. The Tekran® Model 2505 (Tekran, USA) mercury vapour calibration unit is based on the Bell-jar principle. Since the saturation vapour pressure of mercury is a function of temperature, the exact volume injected, and temperature of the mercury saturated air need to be set in order for the Bell-jar to determine the mercury injection amount based on the Dumarey equation (Brown et al., 2008b, Dumarey et al., 1979, Dumarey et al., 1985). Injections, by use of a gas-tight syringe, between 100 µL and 800 µL have been used (including a zero-linearity check point) to calibrate the analyser in the range of 1.3 ng to 10.5 ng.

## 2.4 Second comparison: 10 ng – 1000 ng mercury

Gold sorbent tubes have been used for the 10 ng, 50 ng, 100 ng and 500 ng samples and carbon traps for the 1000 ng samples. Commercially available gold-coated silica (Brooks Rand Instruments, US) and in-house prepared gold-coated high-grade corundum sand (mass fraction gold of 9.6 %) have been used in this study to prepare gold traps. Clean quartz tubes (i.d. 5 mm) were filled with gold-coated silica or corundum (20 mm length) and fixed with quartz wool. Carbon traps were prepared by filling clean quartz tubes (i.d. 5 mm) with 20 mm length of iodinated activated carbon (AIC-500 from Apex Instruments, US) and fixed with quartz wool. For this comparison 3 or 4 sorbent tubes were sampled simultaneously at 5 different levels according to Table 3 using the diffusion cells with a capillary of 3 mm or 33 mm in diameter (diffusion rates of 4.23 ng min$^{-1}$ or 218.5 ng min$^{-1}$ respectively). The air flow through the system was kept constant at 3.00 L min$^{-1}$ ± 0.02 L min$^{-1}$ ($k = 2$) to obtain a mercury concentration of 1.41 µg m$^{-3}$ ± 0.04 µg m$^{-3}$ and 71.4 µg m$^{-3}$ ± 0.9 µg m$^{-3}$ respectively. Other parameters, i.e., the diffusion cells, sample flow and sampling time were changed during the different entries to demonstrate the variability of the primary mercury gas standard set up.

**Table 3: Parameters used to sample sorbent tubes from the primary mercury gas standard for the second comparison**

| Entry | Diffusion flow rate (ng min$^{-1}$) (U ($k = 2$)) | Sample flow rate (L min$^{-1}$) (U ($k = 2$)) | Sampling time (min) (U ($k = 2$)) | Mercury amount per tube (ng) (U ($k = 2$)) |
|---|---|---|---|---|
| 1 | 4.23 (0.13) | 0.1002 (0.0004) | 75.00 (0.03) | 10.5 (0.3) |

| 2 | 4.23 (0.13) | 0.1999 (0.0007) | 37.50 (0.03) | 10.5 (0.3) |
|---|---|---|---|---|
| 3 | 4.23 (0.13) | 0.1999 (0.0007) | 37.50 (0.03) | 10.5 (0.3) |
| 4 | 4.23 (0.13) | 0.4997 (0.0017) | 75.00 (0.03) | 52.6 (1.7) |
| 5 | 4.23 (0.13) | 0.4997 (0.0017) | 150.00 (0.03) | 105 (3) |
| 6 | 214.3 (2.4) | 0.0499 (0.0002) | 14.00 (0.03) | 49.8 (0.7) |
| 7 | 214.3 (2.4) | 0.1003 (0.0004) | 15.00 (0.03) | 107.3 (1.5) |
| 8 | 214.3 (2.4) | 0.4997 (0.0017) | 15.00 (0.03) | 535 (8) |
| 9 | 214.3 (2.4) | 0.5000 (0.0018) | 15.00 (0.03) | 537 (8) |
| 10 | 214.3 (2.4) | 0.5000 (0.0018) | 15.00 (0.03) | 537 (8) |
| 11 | 214.3 (2.4) | 0.5000 (0.0018) | 15.00 (0.03) | 537 (8) |
| 12 | 214.3 (2.4) | 0.5000 (0.0018) | 15.00 (0.03) | 537 (8) |
| 13 | 214.3 (2.4) | 0.5001 (0.0018) | 30.00 (0.03) | 1070 (15) |
| 14 | 214.3 (2.4) | 0.1004 (0.0004) | 150.00 (0.03) | 1074 (15) |

The sorbent tubes were analysed directly after sampling by JSI in the VSL laboratories in November 2018. The amount of mercury on the carbon traps was determined using mercury analyser RA-915M with a PYRO-915+ thermal decomposition attachment (Lumex Scientific, St. Petersburg, Russia), that is based on differential Zeeman atomic absorption spectrometry. Iodinated activated carbon and quartz wool from the carbon trap were quantitatively transferred to a quartz boat and mercury was released from the sample by combustion in the thermal decomposition unit at 700 °C. The system was calibrated by spiking

a known amount of NIST SRM 3133 standard solution (1000 ng) to iodinated activated carbon and determining the corresponding signal using the same procedure as for the sample. The signal of procedural blanks (average value 1.9 ng) was subtracted from the corresponding sample (standard) signal.

The amount of mercury on the gold traps was determined using mercury analyser RA-915M with a modification of the PYRO-915+ to allow direct desorption of mercury from the gold trap in the thermal decomposition unit. The system was calibrated

by reducing a known amount of NIST SRM 3133 standard solution (10 ng – 500 ng) with tin(II) chloride solution, quantitative purging of the obtained $Hg^0$ gas onto a gold trap (EPA Method 1631, 2002), and its desorption in the thermal decomposition unit (Shulupov et al., 2004). The signal of procedural blank was always within the instrumental noise and could not be subtracted from the corresponding sample (standard) signal.

The uncertainty of the analytical procedure was estimated using the law of propagation of uncertainty (BIPM, 2008). Estimated

relative standard uncertainty of the calibration was calculated from the contributions of the NIST SRM 3133 (0.24 %, $k = 1$), uncertainties of pipettes used for spiking and preparation of NIST SRM 3133 dilutions, and the uncertainty of the volume due to possible temperature changes. The estimated relative standard uncertainty of the calibration ranged from 0.5 % to 0.6 % ($k = 1$) for 1000 ng and 10 ng spike, respectively. The relative expanded uncertainty of the whole analytical procedure was usually

2.5 % – 3 % ($k = 2$) and included effects such as sampling repeatability as the greatest contribution, and effects due to calibration and recovery.

## 3. Results

### 3.1 First comparison: 2 ng – 10 ng mercury

For the first comparison Amasil sorbent tubes were sampled in three rounds with amounts between 2 ng and 10 ng of mercury. Two tubes sampled simultaneously per level, as well as blank tubes, were sent to NPL for analysis after each sampling for each of the three rounds. Directly after round 1 three tubes were analysed by VSL for each sampling level. Both VSL and NPL calibrated their analyser with injections from the bell-jar and the mercury amount is calculated using the Dumarey equation. This approach enables determining the difference between the primary mercury gas standard and the Dumarey equation. After calibration of the equipment the mercury amount on the tubes is verified against the Dumarey equation (Figure 1).

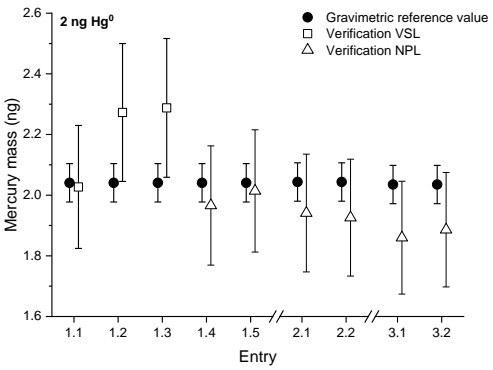
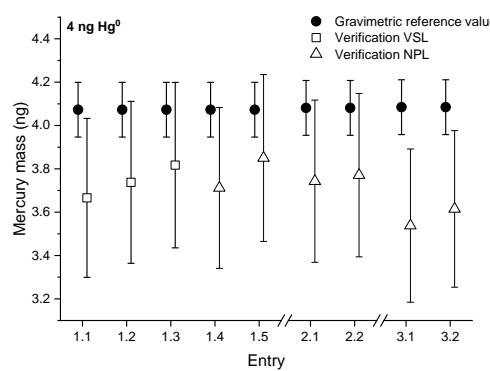
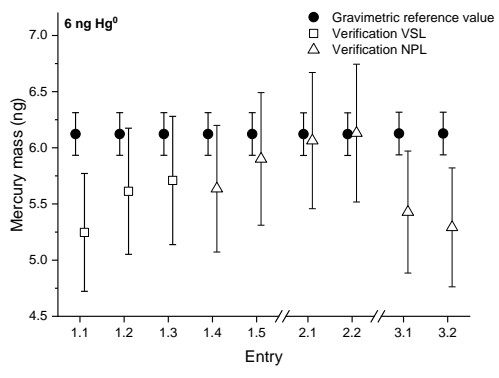
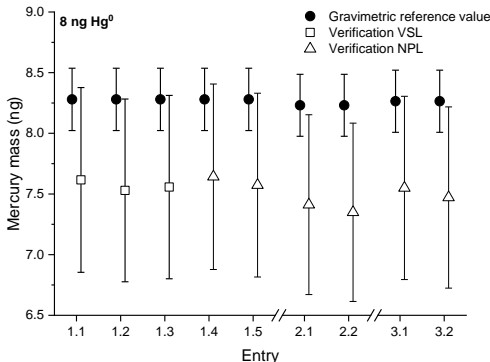

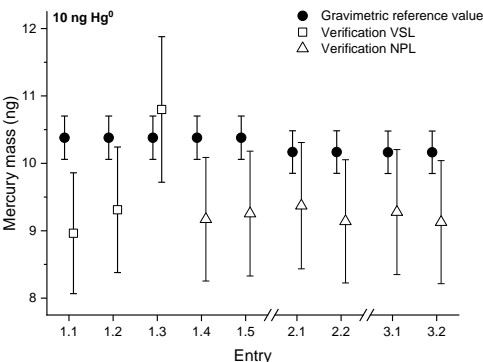

**Figure 1: Verification results of the first comparison with mercury amounts of approximately 2, 4, 6, 8 and 10 ng on sorbent tubes. The open squares and triangles are the results from the measurements calibrated using the Dumarey equation. The closed circles are the reference values determined gravimetrically. The error bars indicate the expanded uncertainty for the reference values (ng) ($k = 2$) and the verification values (ng) ($k = 2$). Entry 1.1, 1.2 and 1.3 show the verification results of VSL from the first round. The verification results of NPL in the first round are entry 1.4 and 1.5, in the second round 2.1 and 2.2 and in the third round 3.1 and 3.2.**

The blank tubes were not sampled with mercury but have been returned to NPL to check for contamination during transport and storage. The recovery of all the blank tubes analysed during the three rounds is below 0.1 ng. The repeatability and reproducibility standard deviation have been calculated according to ISO 5725-2 (ISO 5725-2, 2019). Based on the results of NPL the repeatability standard deviation of the sampling (within rounds) is 2.2 % and the reproducibility standard deviation (over three rounds, with exception of the 6 ng samples) is 3 %. For the 6 ng samples the reproducibility standard deviation is 6.7 %. The reproducibility standard deviation of the verification results is equal to the uncertainty of the mercury amount (Table 1), except for the 6 ng samples. This reproducibility standard deviation is below the uncertainty of the analysis, which is ≤ 10 %. The verification results obtained during round 1 by NPL and VSL are comparable for the 4 ng, 6 ng, 8 ng and 10 ng samples. The difference between the results is well within the uncertainty of the analysis. For the 2 ng samples the result of entry 1.1 is comparable with the NPL verification results, however, the results for entry 1.2 and 1.3 are not (Figure 1). In general, the verification of the samples shows results below the reference value with a few exceptions. The average difference between the reference values and the verification results is -8 % with a standard deviation of 6 %. This implies there is a difference between the primary mercury gas standard and the Dumarey equation of -8 %. Based upon these results, measurement results based upon the Dumarey equation have a -8 % measurement bias.

**3.2 Second comparison: 10 ng – 1000 ng mercury**

During the second comparison 3 or 4 sorbent tubes were sampled simultaneously at VSL and analysed directly at VSL. For the verification the average recovery of the 3 or 4 sorbent tubes is reported (Figure 2). During the verification the analysers are calibrated with the NIST liquid reference material (SRM 3133). For Entry 1 – 5 (10 ng, 50 ng and 100 ng (Table 3)) the diffusion cells with a capillary of 3 mm in diameter have been used. The diffusion cells with a capillary of 33 mm in diameter

have been used for sampling the other Entries (Entry 6 – 14; 50 ng, 100 ng, 500 ng and 1000 ng (Table 3)). As such, for sampling the sorbent tubes with 50 ng and 100 ng, both types of diffusion cells have been used.

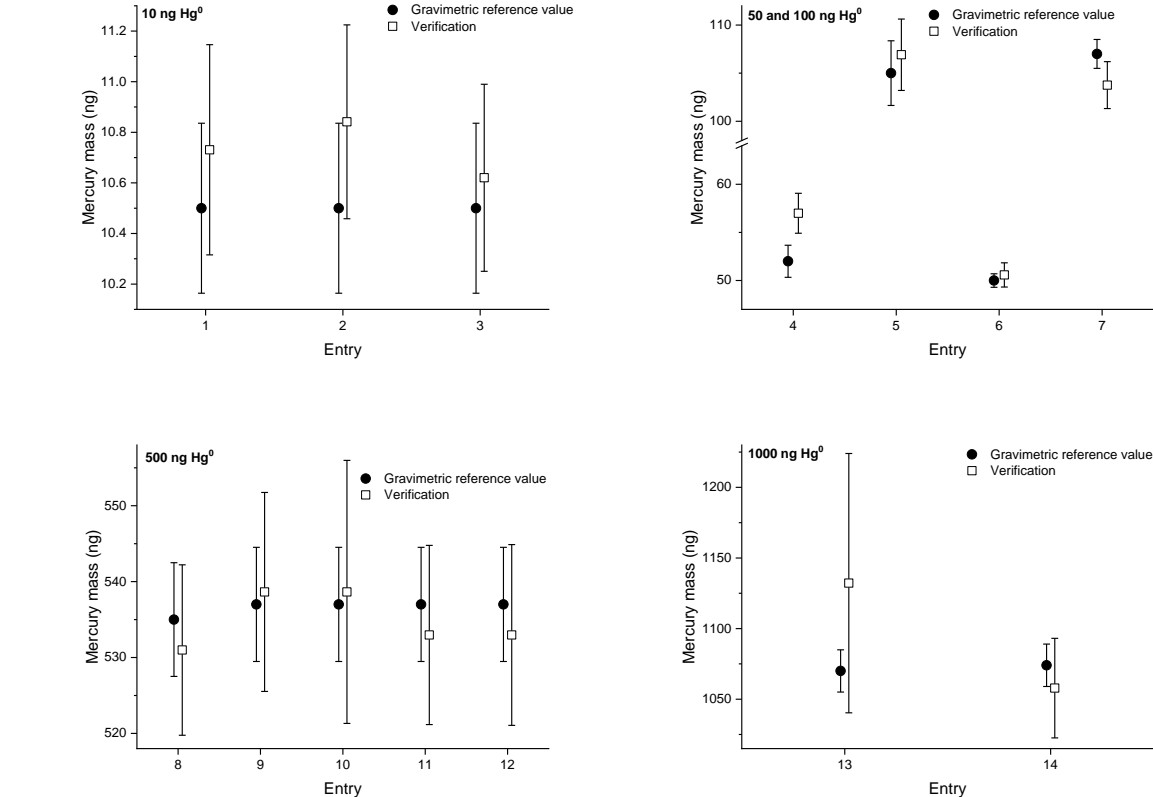

**Figure 2: Verification results of the second comparison with mercury amounts of approximately 10, 50, 100, 500 and 1000 ng on sorbent tubes. The open squares are the verification results. The closed circles are the reference values determined gravimetrically. The error bars indicate the expanded uncertainty for the reference value (ng) ($k = 2$) and the verification value (ng) ($k = 2$).**

The repeatability and reproducibility standard deviation of the 10 ng and 500 ng samples have been calculated according to ISO 5725-2 (ISO 5725-2, 2019). Repeatability standard deviations of 0.8 % for both levels have been obtained and the reproducibility standard deviation is 1.3 % and 0.9 % for the 10 ng and 500 ng samples, respectively. For the samples obtained using the diffusion cells with a capillary of 3 mm in diameter an average difference, between the reference values and the verification results, of +3.6 % has been obtained, except for Entry 4. In that case the reference value and verification

result do not overlap, with a difference of +9.6 %. For the samples obtained using the diffusion cells with a capillary of 33 mm in diameter the difference between the reference values and the verification results is very small, just +0.1 %. The average difference between all the reference values and verification results is +1.3 % with a standard deviation of 3 %.

## 4. Discussion

In contrast to the primary mercury gas standard described in this work (based on diffusion according to ISO 6145-8 (ISO 6145-8, 2005)) gaseous elemental mercury generators available on the market are based on the dilution of a saturated mercury atmosphere to obtain mercury concentrations according to ISO 6145-9 (ISO 6145-9, 2009). For many years different vapour pressure-temperature relationships for the computation of the output of such generators have been described and compared. Approximately 30 years ago the Dumarey equation was established which corresponds to the least-squares best fit of results obtained for measurements of mercury mass concentration in air at saturation (Dumarey et al., 2010).

Huber et al. described a correlation based on fitting a thermodynamically constrained model over a wide temperature range to numerous mercury vapour pressures (Huber et al., 2006). However, at room temperature (22 °C) the Dumarey equation yields a value which is 6.4 % lower compared to the Huber equation.

More recently results using SI-traceable mass spectroscopy to determine gaseous elemental mercury concentrations have been reported by Quétel et al., and Srivastava and Hodges. Quétel et al. reported mercury concentration values nearly 3 % and 10 % higher compared to predictions (at 20 °C) based on the Huber equation and Dumarey correlations, respectively (Quétel et al., 2016). The comparison performed by Srivastava and Hodges showed that their measurement results are equivalent within their experimental uncertainty to the Huber equation and Quétel method (Srivastava and Hodges, 2018). Furthermore, the mercury vapour pressure value predicted from the Dumarey equation is 8.5 % below the value based on the results from Srivastava and Hodges.

Based on the results from the two comparisons described in this paper similar conclusions can be drawn. The primary mercury gas standard and the NIST SRM 3133 are comparable within 1.3 %. In contrast the output of the Dumarey equation, used in the first comparison, shows a difference of approximately -8 % compared to the primary mercury gas standard. This discrepancy is outside the uncertainty range for the Dumarey equation (4 %; $k = 2$) and the primary gas standard (3 %; $k = 2$).

In extension to the comparisons highlighted in this work, future intercomparisons between the primary mercury gas standard and the primary measurement method based on Laser Absorption Spectroscopy (LAS) (Srivastava et al., 2020) are planned.

## 5. Conclusion

The work described in this paper describes the comparison between mercury calibration methods maintained by two National Metrology Institutes (NMIs) and a Designated Institute (DI), using the bell-jar in combination with the Dumarey equation or NIST SRM 3133 for the calibration of the equipment. The results of the comparisons show that robust mercury transfer standards can be obtained via sampling of the primary mercury gas standard with a reproducibility standard deviation 3 %, which is equal to the uncertainty of the mercury amount sampled onto the sorbent materials. Transfer standards containing gold or carbon as sorbent material can be prepared with levels between 2 ng and 1000 ng with a relative expanded uncertainty 3 %.

Based on the method used for the calibration of the analyser a difference of -8 % has been obtained between the primary mercury gas standard and the Dumarey equation and a difference of +1.3 % when using NIST SRM 3133 to calibrate the analysers. This implies that the primary mercury gas standard and NIST SRM 3133 are comparable within their measurement uncertainty, whereas a difference of approximately 8 % exists between measurement results based on these two versus the output of the Dumarey equation. Based upon this work it should be emphasized that measurement results based on the Dumarey

equation have a negative bias of approximately 8 %.

The transfer standards obtained with the primary mercury gas standard proved to be useful to establish metrological traceability to the SI units and can be used for the calibration of mercury equipment used in the field, e.g. analysers, bell-jars and gas generators. Furthermore, such transfer standards can be used to benchmark the results of laboratories involved in mercury measurements, e.g. by proficiency tests. The obtained results enable traceable mercury measurement results in emission

sources and the atmosphere. These measurements are of fundamental importance to reduce the mercury burden on the environment, to comply with related regulation and protect human health.

**Acknowledgment**

This project (16ENV01 MercOx) has received funding from the EMPIR programme co-financed by the Participating States and from the European Union's Horizon 2020 research and innovation programme. Warren Corns is acknowledged for

providing iodinated activated carbon.

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
