# Peer review of "Comparability of calibration strategies for measuring mercury concentrations in gas emission sources and the atmosphere"

_Atmospheric Measurement Techniques, 2020_

## Referee Comment (RC1) · Anonymous Referee #1 · 13 Dec 2020

The topic of this work is timely and within the scope of AMT. Mercury is an important pollutant and it is important to have accurate methods of the determination of its concentration in the atmosphere. This work is well-prepared and contains an adequate uncertainty analysis. The comparisons presented are important since they demonstrate that methodologies that incorporate the Dumarey equation are subject to a significant negative bias. This leads to increased amounts of mercury being released into the environment, and more accurate methods should be adopted to replace less reliable, outdated methods.

---

## Referee Comment (RC2) · Seth Lyman (Referee) · 16 Dec 2020

This is a great paper. Very well written, very clear and precise. Also, the results are very important and timely. I have a few specific comments:

Line 73: I know this device is described in the listed references, but would be better to include more description in this text, also, I think. Something just a little more detailed than lines 75-80.

Lines 81-85: At what flow rate(s)?

Figure 1 is a bit confusing to me. So the open squares and open triangles are the

results from the measurements calibrated with the Dumarey equation, right? And the closed circles are determined gravimetrically? Maybe it would be better to label them as such, or at least explain it in the figure caption. I was confused that the closed circles are labled "reference value 2 ng" when they don't have a mass of 2 ng. I know all this is explained in the text, but your paper would make its point better if the figures could stand on their own.

In the conclusions, I think it would be better if you added a paragraph to talk about how others have drawn similar conclusions. You do this briefly on lines 224-226, but a more thorough discussion in the conclusions section would be better. In addition to references in lines 224-226, you should also reference and discuss results obtained by Srivastava and Hodges (2018) and Quétel et al. (2016).

---

## Short Comment (SC1) · 22 Dec 2020

Thank you for reviewing our article and your positive comment. With our work the accuracy and traceability of mercury measurement data can be improved at atmospheric measurement sites and gas emission sources.

---

## Author Comment (AC1) · 5 Jan 2021

Dear Seth Lyman,

Thank you for reviewing our article and your positive and good comments. Below the response to your comments. Line 73: I know this device is described in the listed references, but would be better to include more description in this text, also, I think. Something just a little more detailed than lines 75-80. Lines 81-85: At what flow rate(s)? Indeed, the description is very brief, the text will be improved. Below the more detailed description including the flow rate(s) used in Line 81-85.

The primary gas standard has been developed as an elemental mercury (Hg0) gas generator to establish metrological traceability of mercury concentration measurement results, based upon a gravimetric approach, for ambient air levels as well as higher concentrations (Ent et al., 2014, de Krom et al., 2020). The working principle of the primary mercury vapour generator is based on diffusion according to ISO 6145-8 (ISO 6145-8, 2005). Specially developed stainless-steel diffusion cells are filled with approximately 2 mL of Hg0. The Hg0 vapour diffuses from the cell through a capillary. To obtain several diffusion ranges, cells with different capillary diameters are used, e.g.: 33 mm and 3 mm. By weighing the diffusion cells at regular time intervals on a high-resolution analytical balance (AX1006 mass comparator with a Mettler AT1005 balance, Mettler, Switzerland) the Hg0 mass flow rate (diffusion) is determined gravimetrically. The characterisation of different diffusion cells has been described previously by I. de Krom et al (de Krom et al., 2020). In the dynamic mercury gas generator, the cells are housed in a diffusion chamber. The diffusion chamber is temperature (20.0 oC ± 0.1 oC) and pressure (105.0 kPa ± 0.1 kPa) controlled, At the bottom a nitrogen flow enters the diffusion chamber. All of the flow, also enriched by Hg0 vapour, is then guided out of the diffusion chamber through an aperture at the top. Standard Hg0 gas mixture concentrations are prepared by mixing the Hg0 vapours in nitrogen with flows, between 1 L min-1 and 15 L min-1, of matrix gas, e.g., purified air. Using diffusion cells with a capillary of 3 mm in diameter, mercury concentrations between 0.1 $\mu$g m-3 and 2.1 $\mu$g m-3 can be obtained with an expanded uncertainty of 3 %. Diffusion cells with a capillary of 33 mm in diameter can generate mercury concentrations between 5 $\mu$g m-3 and 100 $\mu$g m-3 with an uncertainty of 1.8 %. In this project both types of diffusion cell have been used to obtain the primary mercury gas standard.

Figure 1 is a bit confusing to me. So the open squares and open triangles are the results from the measurements calibrated with the Dumarey equation, right? And the closed circles are determined gravimetrically? Maybe it would be better to label them as such, or at least explain it in the figure caption. I was confused that the closed circles are labled "reference value 2 ng" when they don't have a mass of 2 ng. I know all this

is explained in the text, but your paper would make its point better if the figures could stand on their own. The figure indeed displays what you explain. It would be good if the figure can stand on its own therefore the figure and capitation have been updated accordingly, see Figure 1. Also figure 2 has been improved accordingly.

In the conclusions, I think it would be better if you added a paragraph to talk about how others have drawn similar conclusions. You do this briefly on lines 224-226, but a more thorough discussion in the conclusions section would be better. In addition to references in lines 224-226, you should also reference and discuss results obtained by Srivastava and Hodges (2018) and Quétel et al. (2016).

Thank you for your comment, an extra chapter "Discussion" has been added in which the results in former publications are disused and the references have been included. In contrast to the primary mercury gas standard described in this work (based on diffusion according to ISO 6145-8 (ISO 6145-8, 2005)) gaseous elemental mercury generators available on the market are based on the dilution of a saturated mercury atmosphere to obtain mercury concentrations according to ISO 6145-9 (ISO 6145-9, 2009). For many years different vapour pressure-temperature relationships for the computation of the output of such generators have been described and compared. Approximately 30 years ago the Dumarey equation was established which corresponds to the least-squares best fit of results obtained for measurements of mercury mass concentration in air at saturation (Dumarey et al., 2010). Huber et al. described a correlation based on fitting a thermodynamically constrained model over a wide temperature range to numerous mercury vapour pressures (Huber et al., 2006). However, at room temperature (22 oC) the Dumarey equation yields a value which is 6.4 % lower compared to the Huber equation. More recently results using SI-traceable mass spectroscopy to determine gaseous elemental mercury concentrations have been reported by Quétel et al., and Srivastava and Hodges. Quétel et al. reported mercury concentration values nearly 3 % and 10 % higher compared to predictions (at 20 oC) based on the Huber equation and Dumarey correlations, respectively (Quétel et al., 2016). The comparison

performed by Srivastava and Hodges showed that their measurement results are equivalent within their experimental uncertainty to the Huber equation and Quétel method (Srivastava and Hodges, 2018). Furthermore, the mercury vapour pressure value predicted from the Dumarey equation is 8.5 % below the value based on the results from Srivastava and Hodges. Based on the results from the two comparisons described in this paper similar conclusions can be drawn. The primary mercury gas standard and the NIST SRM 3133 are comparable within 1.3 %. In contrast the output of the Dumarey equation, used in the first comparison, shows a difference of approximately -8 % compared to the primary mercury gas standard. This discrepancy is outside the uncertainty range for the Dumarey equation (4 %; k = 2) and the primary standard (3 %; k = 2). In extension to the comparisons highlighted in this work, future intercomparisons between the primary mercury gas standard and the primary measurement method based on Laser Absorption Spectroscopy (LAS) (Srivastava et al., 2020) are planned.

[Figure]

**Figure 1: Verification results of the first comparison with mercury amounts of approximately 2, 4, 6, 8 and 10 ng on sorbent tubes. The open squares and triangles are the results from the measurements calibrated with the Dumarey equation. The closed circles are the reference values determined gravimetrically. The error bars indicate the expanded uncertainty for the reference values (ng) ($k$ = 2) and the verification values (ng) ($k$ = 2). Entry 1.1, 1.2 and 1.3 show the verification results of VSL from the first round. The verification results of NPL in the first round are entry 1.4 and 1.5, in the second round 2.1 and 2.2 and in the third round 3.1 and 3.2.**

**Fig. 1.**

[Figure]

**Figure 2: Verification results of the second comparison with mercury amounts of approximately 10, 50, 100, 500 and 1000 ng on sorbent tubes. The open squires are the verification results. The closed circles are the reference values determined gravimetrically. The error bars indicate the expanded uncertainty for the reference value (ng) ($k$ = 2) and the verification value (ng) ($k$ = 2).**

**Fig. 2.**

---

## Author Comment (AC2) · 13 Jan 2021

Thank you for reviewing our article and your positive comment. With our work the accuracy and traceability of mercury measurement data can be improved at atmospheric measurement sites and gas emission sources.